# Effects of Global Warming on Patients with Dementia, Motor Neuron or Parkinson’s Diseases: A Comparison among Cortical and Subcortical Disorders

**DOI:** 10.3390/ijerph192013429

**Published:** 2022-10-18

**Authors:** Paolo Bongioanni, Renata Del Carratore, Cristina Dolciotti, Andrea Diana, Roberto Buizza

**Affiliations:** 1Medical Specialties Department, Azienda Ospedaliero-Universitaria Pisana, 56100 Pisa, Italy; 2NeuroCare onlus, 56100 Pisa, Italy; 3Institute of Clinical Physiology, National Research Council, 56100 Pisa, Italy; 4Department of Biomedical Sciences, University of Cagliari, 09100 Cagliari, Italy; 5Life Science Institute, Scuola Superiore Sant’Anna, 56100 Pisa, Italy

**Keywords:** global warming, dementia, motor neuron disease, Parkinson’s disease, neurodegeneration

## Abstract

Exposure to global warming can be dangerous for health and can lead to an increase in the prevalence of neurological diseases worldwide. Such an effect is more evident in populations that are less prepared to cope with enhanced environmental temperatures. In this work, we extend our previous research on the link between climate change and Parkinson’s disease (PD) to also include Alzheimer’s Disease and other Dementias (AD/D) and Amyotrophic Lateral Sclerosis/Motor Neuron Diseases (ALS/MND). One hundred and eighty-four world countries were clustered into four groups according to their climate indices (warming and annual average temperature). Variations between 1990 and 2016 in the diseases’ indices (prevalence, deaths, and disability-adjusted life years) and climate indices for the four clusters were analyzed. Unlike our previous work on PD, we did not find any significant correlation between warming and epidemiological indices for AD/D and ALS/MND patients. A significantly lower increment in prevalence in countries with higher temperatures was found for ALS/MND patients. It can be argued that the discordant findings between AD/D or ALS/MND and PD might be related to the different features of the neuronal types involved and the pathophysiology of thermoregulation. The neurons of AD/D and ALS/MND patients are less vulnerable to heat-related degeneration effects than PD patients. PD patients’ substantia nigra pars compacta (SNpc), which are constitutively frailer due to their morphology and function, fall down under an overwhelming oxidative stress caused by climate warming.

## 1. Introduction

Climate change represents a global phenomenon [1], extensively disrupting ecosystems worldwide, that can have a dramatic impact on human health [2]. An increased prevalence of neurological diseases, also enclosing neurodegenerative ones, such as Alzheimer’s Disease and other Dementias (AD/D), Amyotrophic Lateral Sclerosis/Motor Neuron Diseases (ALS/MND), and Parkinson’s Disease (PD) has been reported [3]. We have recently reviewed scientific works that discuss the impact of climate change induced by global warming and in particular the increased frequency and intensity of heat waves, on health problems. Global warming can significantly increase the rate of neurodegenerative disorders by inducing persistent heat stress even at temperature ranges that, acutely, would be tolerable for neurons [4].

Although an increase in the prevalence of neurodegenerative diseases is well documented, the link between global warming and the enhanced prevalence of such diseases remains elusive. Heat stress due to global warming could significantly increase the rate of neurodegenerative disorders [3,5,6,7] by causing DNA damage, protein misfolding and aggregation, microglia activation, neuroinflammation, apoptosis, and autophagy induction within neurons [8,9] which could further expose them to degeneration (Figure 1).

In order to investigate whether there is any evidence of the effect of global warming on neurodegenerative diseases, we have recently studied the link with PD epidemiology by contrasting variations between 1990 and 2016 in the epidemiological indices of PD patients (deaths, prevalence, and disability-adjusted life years—DALYs) and the climate indices of 185 countries in the world [10]. In the cluster that includes the 25 countries (corresponding to about 900 million people), which were characterized by a higher-than-average warming and higher-than-average temperature, we found a positive correlation of about 25% between more intense warming and higher variations in the PD indices. We concluded that this should be considered as statistically significant, providing evidence that climate change is one of the key environmental factors that can influence human health.

Here, we expanded our investigation to two other neurodegenerative diseases AD/D and ALS/MND which as well as PD, are neuropathologically characterized by protein misfolding and aggregation into extracellular deposits. As in Buizza et al. [10], the same countries are considered, and variations of disease indices and climate indices between 1990 and 2016 have been contrasted and statistically analyzed for 4 clusters of countries.

## 2. Methods

### 2.1. Data Collection

Epidemiological data of AD/D and ALS/MND were derived, respectively, from Nichols et al. [11] and Logroscino et al. [12], who listed the global, regional, and country-specific variations between 1990 and 2016 of 3 epidemiological indices (prevalence, deaths, and DALYs) related to AD/D and ALS/MND patients. PD epidemiological data were derived from Dorsey et al. [13].

Climate data were extracted from the World Bank Climate Change Data Portal (WBCCDP—freely available from The World Bank website: https://data.worldbank.org/indicator, accessed on 1 September 2021) and included, for almost all the countries of the world, the monthly average temperature from 1990 to 2016. From these data, two climate indices were defined [10]:-The average temperature in 2016 (T2016) was computed as the annual mean 2 m temperature. The annual temperature in 2016 was used simply to cluster the countries in ‘warmer than the median’ or ‘colder than the median’ categories. Using the average temperature of 1990 (T1990) instead of T2016 would not change the conclusion of the work since a very small number of countries would move from one cluster to the other if T1990 was used instead of T2016. For each country, T2016 was computed as the average of the monthly temperatures of 2016 available in the data set:
$$T2016=\frac{1}{12}\overset{12}{\underset{i=1}{\sum }}\left(T2016_{\mathrm{JAN}}+T2016_{\mathrm{MAR}}+\ldots +T2016_{\mathrm{DEC}}\right)$$

The warming index was computed by considering all the monthly temperatures from the years 1990 to 2016 before fitting a linear regression line and setting the warming index to be equal to the slope of the linear fit line. More precisely, when one country was considered, all months from January 1990 to December 2016 were identified with the index m = 1324 and denoted with Tm, the country’s monthly average temperature for month m. If the country experienced warming due to climate change, the linear-fit curve, which explains the trend of the monthly average temperatures from 1990 to 2016, had a positive slope: Tm = am + b, where a is the slope of the linear fit, and b is the intercept. The country warming index was defined as the slope of this straight line: WI1990–2016 = a.

-The 1990–2016 warming index (WI1990–2016) can be seen in the slope of the linear regression curve, which fits the monthly 2 m temperature data from January 1990 to December 2016.

According to these indices, countries with a warmer climate have a higher value of T2016, and countries more affected by climate change have a higher WI1990–2016.

### 2.2. Subjects

Of the more than 190 countries listed in Nichols et al. [11], Logroscino et al. [12], and Dorsey et al. [13], we selected the 184 for which we could find epidemiological data for all 3 diseases and monthly average temperature data (Appendix A). These countries, once their climate indices were computed, were grouped in 4 clusters (Appendix A) by comparing their individual climate indices with the sample medians: the high-temperature and high-warming (HT-HW) cluster; the high-temperature and low-warming (HT-LW) cluster; the low-temperature and high-warming (LT-HW) cluster; and the low-temperature and low-warming (LT-LW) cluster. We chose to apply the simplest possible way to cluster the countries, warmer and colder, and in the ones that warmed more or less during the 27 years. This simple method was inspired by the fact that we expected people living in warm countries that experienced the strongest warming to show a stronger impact of climate change.

### 2.3. Statistical Analyses

The characteristics of the distributions of the 4 clusters have been described by their means, medians and quartiles, and standard statistical methods have been applied [10] to assess the correlation between variables. When comparing two distributions to assess whether they came from the same overall population, Student’s *t*-tests were computed. Student *t*-tests provided the statistical significance and the validity of the null-hypothesis that is posed throughout the analysis. A Student’s *t*-test *p*-value less than 5% would indicate that the distributions of prevalent variations in the two clusters were different. The null hypothesis that ‘the two sampled distributions come from the same overall population’ can be rejected at the 5% level (i.e., there is a 95% probability that they come from a different overall population).

## 3. Results

### 3.1. Climate Indices

There is a huge variation between the two climate indices within the 184 countries: T2016 varies between 15.9 °C and 29.1 °C, with a median of 23.3 °C, and WI1990–2016 varies between 0.3 °C and 1.8 °C, with a median of 0.7 °C. The numbers quoted are the minimum and maximum values. The scatter plot of the two climate indices, with the countries clustered in the four categories introduced above, is shown in Figure 2, which indicates that colder countries tended to have a higher warming index than warmer countries, and that the distributions of T2016 was skewed towards colder values (skewness equal to 1.02).

The skewness of the two distributions explains why the number of countries in the HT-HW and the LT-LW clusters is much smaller than the number of countries in the other clusters (cluster HT-HW includes 25 countries; cluster HT-LW includes 68; cluster LT-HW includes 67, and cluster LT-LW includes 24 countries).

### 3.2. Epidemiological Indices Related to Country Clusters

To investigate the link between climate and neurodegenerative diseases, a statistical analysis was performed by correlating the epidemiological with the climate indices. Data referring to prevalence are reported in Table 1 and Figure 3. Data referring to deaths and DALYs are reported in Appendix A. The median values of the prevalence variations during the 1990–2016 period for the studied neurodegenerative diseases are reported in Table 1. The *p*-values of the Student’s *t*-test calculated between the distributions of prevalence for the different clusters (HT-HW, HT-LW, LT-HW, and LT-LW) are reported in Table 2.

The scatter plots for the prevalence and WI1990–2016 of the countries in the HT-HW cluster are shown for all three diseases: each panel also includes a linear-fit curve, the slope of which shows whether there is any linear relationship between prevalence and climate warming (Figure 3). In ALS/MND patients, the distributions of the two HT clusters differed from the distributions of the two LT clusters (Table 2B), but there was no indication of any sensitivity to climate warming. A significantly lower increment in prevalence was observed in people living in HT-HW (0.044) and HT-LW (0.062) countries as compared to subjects of LT-HW (0.065) or LT-LW (0.071) countries.

Clear significant differences between the median value of the HT-HW cluster and the medians of the other three clusters were observed only for PD patients (Table 1 and Table 2C). The countries in the HT-HW cluster behaved differently than the others and had a higher median value (see Buizza et al. [10] for a more detailed discussion of the PD results). Contrasting results were found for AD/D with only one *p*-value below 5% (Table 2A). Different behaviors between the AD/D and ALS/MND patients and the PD patients could be detected, albeit less clearly, by considering deaths, or DALYs, instead of prevalence as in the epidemiological index (Appendix A).

A linear relationship between the 1990–2016 variations of prevalence and the WI1990–2016 was found for the HT-HW cluster only (Figure 3): the linear-fit did not have any significant slope for AD/D and ALS/MND patients, whereas for PD patients, there was a linear relation with a clear slope and a correlation coefficient of 25% (Figure 3, panel C). This indicates that for PD patients only, living in a warmer-than-average-climate resulted in global warming influencing epidemiological indices. This analysis confirms PD patients’ sensitivity to climate change (Table 1 and Table 2) and also indicates a sensitivity to the underlying climate conditions for ALS/MND patients.

## 4. Discussion

We investigated whether a correlation between climate warming and the variations in epidemiological indices of neurodegenerative diseases between 1990 and 2016 could also be identified for AD/D and ALS/MND patients, as it was found for PD patients [10]. No significant correlation has been found for AD/D or ALS/MND patients. On the other hand, a lower increment in ALS/MND prevalence was found in the HT as compared to the LT clusters indicating that, for these patients, a higher temperature could somehow protect from the disease, as confirmed by the positive effect of observed sunlight exposure on reduced disease incidence in the Taiwan population, regardless of environmental warming [14]. Anyway, the higher climate temperature and/or a progressive environmental warming are reported to lead directly or indirectly to the enhancement of oxidative stress, changes in cerebrovascular hemodynamics, excitotoxicity, and microglial activation, which are all factors implicated in ALS/MND pathogenesis [15]. Epidemiological studies suggest that in ALS/MND, the effects of a prolonged exposure to climate and environment factors could be modulated by the presence of a genetic predisposition, although the interplay between the effect of climate changes and gene-environment interactions still remain unclear [16].

Moreover, climate change is causing a growing population displacement [17] that might explain, by transferring people with a higher susceptibility of ALS/MND from HT to LT countries, the lower disease prevalence in the former as compared to the latter ones [18].

Our data showed no effect of global warming on the epidemiological indices of both AD/D or ALS/MND patients, at least as far the 1990–2016 period is concerned, thereby suggesting that neurons from such patients are less prone to climate warming deleterious effects than those from PD patients, on the basis of the prevailing subcortical rather than cortical localization of the neurodegenerative processes occurring in PD, namely in the substantia nigra pars compacta (SNpc) dopaminergic (DA) neurons, as compared to what happens in AD/D or ALS/MND patients (Figure 4).

The significant reduction in Nogo-A signaling (involved in neuronal dysfunction and repair mechanisms in neurodegenerative disorders) in SNpc DA neurons with an increasing age [19], compared to other neurodegenerative diseases such as AD/D and ALS/MND, are consistent with the concept of higher SNpc DA neuron vulnerability. The scientific literature reports a selectively higher structural, functional and biochemical susceptibility of SNpc DA neurons to degeneration (Figure 4) due to long and highly branched axons (exceeding 4 m in length). DA neurons have a very large number (more than 1 million per cell) of synaptic contacts [20,21] and retain a relatively low cytosolic Ca^++^ buffering capacity due to low levels of calbindin [22,23]. DA neurons might be related to mitochondrial functioning impairments since PD patients are endowed with SNpc DA neurons which are affected by increased mitochondrial DNA deletions when compared with age-matched controls and subjects suffering from other neurodegenerative disorders [24]. Abnormal mitochondrial proteins and the impaired function of the respiratory chain are constant findings in PD patients, as consistently confirmed by the reported decreased levels of prohibitin (a protein localized in the inner mitochondrial membrane and related to mitochondrial stress) in PD patients’ SNpc versus age-matched AD patients and healthy controls [25]. We speculate that although increases in oxidative stress in PD may be widespread in the brain and tolerated by most neurons, in SNpc DA neurons already subjected to elevated oxidative stress from intrinsic sources, a further increase overwhelms antioxidant defenses and leads to degeneration [26].

Another meaningful explanation for the apparently lower heat susceptibility was observed in AD/D or ALS/MND patients compared with PD subjects. In the latter, thermoregulation is earlier and more altered than in patients with other neurodegenerative diseases with predominant cortical involvement [27], making it intriguing to speculate about a possible minor sensitivity to the effects of climate warming.

By considering the above-collected evidence on the cardinal role of mitochondria in their different molecular components, a crucial relevant issue to decrypt the real impact of global warming is linked to a possible dysregulation affecting the adipose tissue (AT). In fact, this is a paramount regulator of energy expenditure in the human body by means of mitochondria intervention. Typically, AT is made of white and brown adipose tissues (BAT) with the former deputed to store energy in the form of triglyceride, and the latter devoted to energy combustion through uncoupling protein-1 (UCP1)-mediated heat production, thereby maintaining body temperature [28]. This is precisely how the regulation of the whole-body energy balance and related body temperature is accomplished [29].

The specific link between BAT and PD has been confirmed by the 6-OHDA-induced PD rat model, where an up-regulation found in the expression of BAT UCP1 was also associated with an enhancement of sympathetic nerve (SN) activity and thermogenesis in PD rats [30,31]. As a matter of fact, BAT is highly innervated by noradrenergic fibers and has numerous blood capillaries: the release of norepinephrine from the terminal of SN triggers the 3-adrenergic receptor binding in BAT, which elicits a downstream cascade of events resulting in lipolysis activation and thermogenesis promotion [32]. These works suggest that weight loss in PD is related to enhanced BAT-mediated thermogenesis [33].

On the other hand, no similar observations have been reported for AD/D and ALS/MND patients and, indeed, a definitive consensus cannot be drawn about the impact of BAT UCP1 dysfunctions on climate warming and dysregulated thermogenesis in such neurodegenerative diseases [34]. In addition, the findings that the hypothalamus—the main thermoregulator under the control of AT [35]— has a more pronounced and earlier degeneration in PD than in other neurodegenerative diseases showing mainly cortical neurodegeneration [36,37] might explain the higher vulnerability of PD patients to climate warming affecting thermoregulation. Interestingly, recent research has found that UCP1 is not restricted to thermogenic AT but also the brain with significant localization in the hypothalamic regions [38].

The dramatic heat variations or global mean temperature rises in some geographical areas might act as a boosting element that further drives hypothalamic neurons to oxidative stress and their final demise by means of the fatal compensative recruitment of UCP1-containing cells. Moreover, in comparison to AD/D and ALS/MND, the dysautonomic syndrome in PD patients, which is clinically reported to be quite early across disease progression and likewise correlated to misfolded α-Synuclein, aggregates in peripheral autonomic neurons [39], exacerbating the already deranged thermoregulation. One of the limitations of this study is that the time frame covered by the disease data we had access to is only 27 years. Considering that the global average temperature has been increasing by about 0.2 °C every 10 years, the global average temperature has increased by about 0.5 °C within 27 years, and only a few countries have experienced warming levels higher than this amount. If we had access to disease data encompassing a longer time span (at least 40 years), we could have possibly found more clear and robust signals.

A second limitation of this study is that, for each country, we only had access to two diseases data points: the first in 1990 and the second in 2016. On the other hand, having the chance to obtained annual data, we could have explored months with extremely hot weather conditions also characterized by higher diseases indices. Some of the recent literature indicates that there could be other important correlations between global warming and neurodegenerative diseases such as AD [7].

## 5. Conclusions

Our findings, which identified significant correlations between epidemiological and climate data in PD patients and no significant correlations for AD/D and ALS/MND patients, could be explained by the fact that AD/D and ALS/MND patients’ neurons are less vulnerable to heat-related degeneration effects than PD patients. PD patients’ SNpc DA neurons, which are constitutively frailer due to their morphology and function, fall down under overwhelming oxidative stress caused by climate warming.

## Figures and Tables

**Figure 1 ijerph-19-13429-f001:**
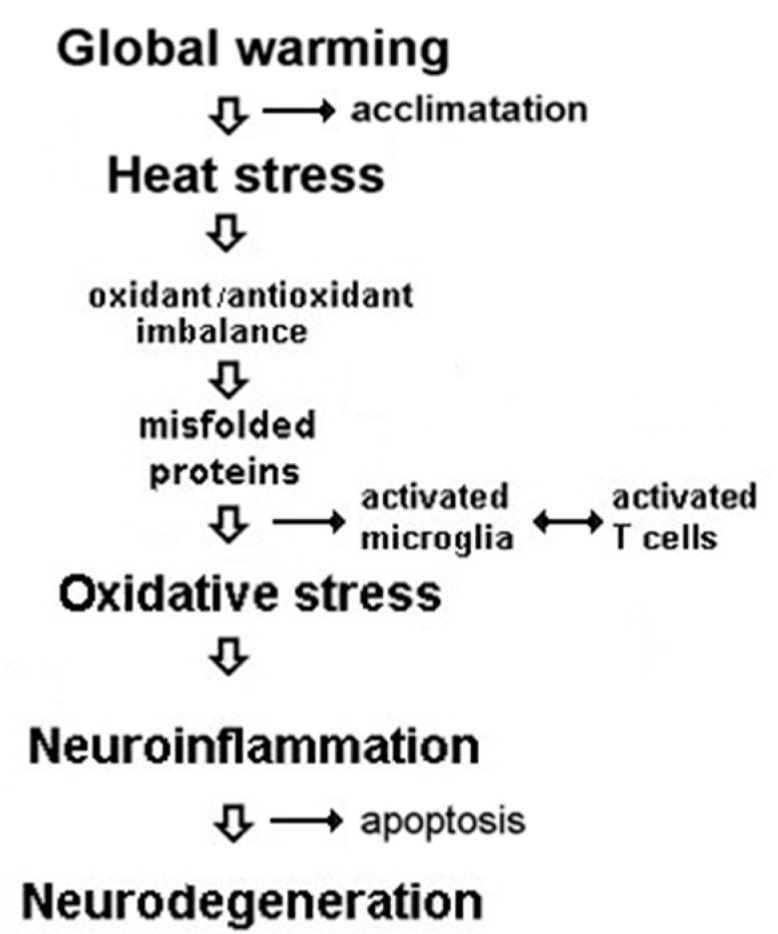
Graphical representation of the main events related to global warming and the possibility of occurrence of various neurodegenerative disorders.

**Figure 2 ijerph-19-13429-f002:**
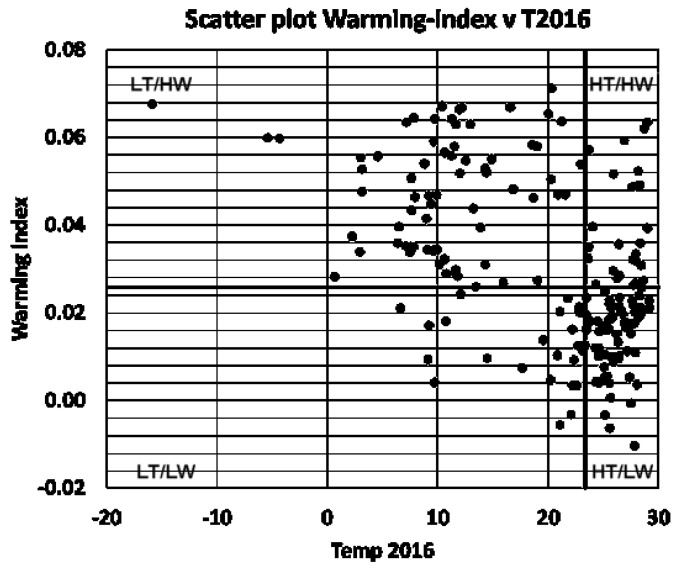
Scatter plot of the 1990−2016 warming index versus T2016. Each symbol represents one of the 184 analyzed countries (Appendix A). The horizontal and vertical lines divide the countries in the four categories high/low temperature, high/low warming.

**Figure 3 ijerph-19-13429-f003:**
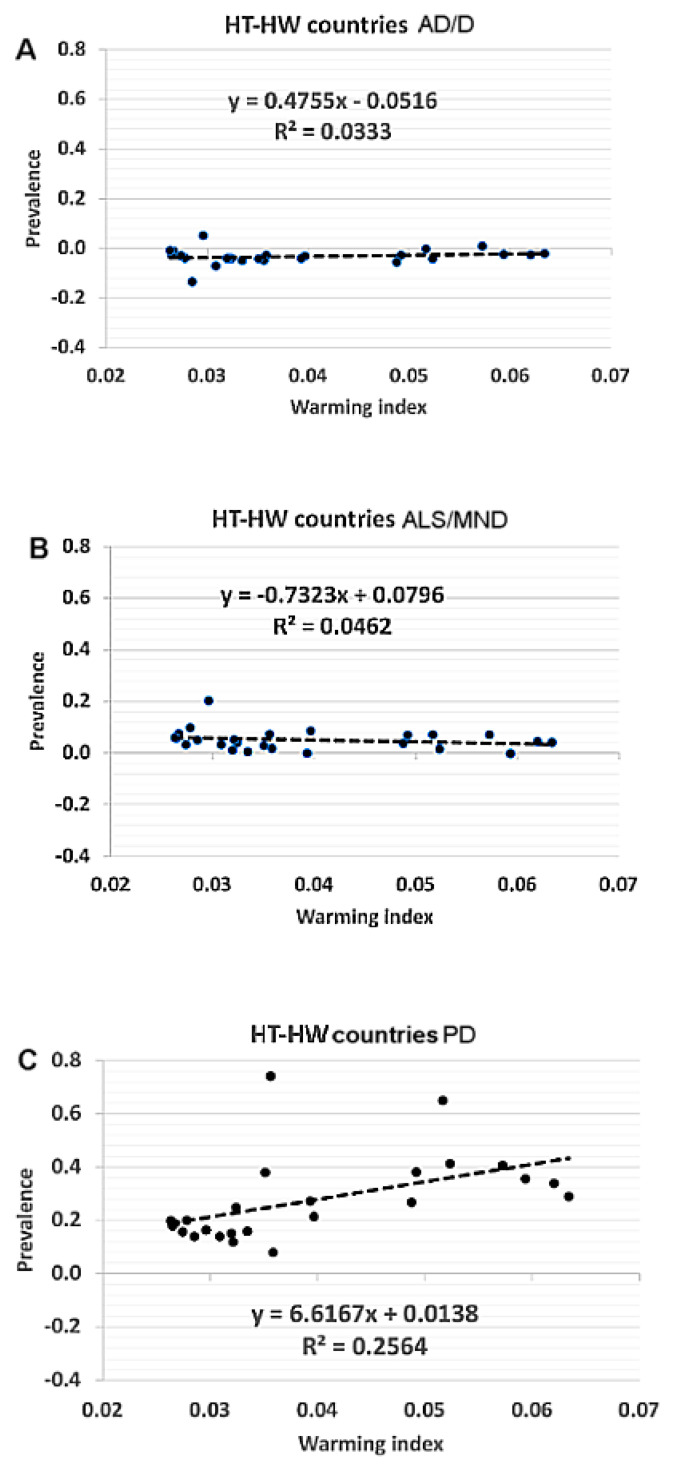
Scatter plots of prevalence and the warming index. Relationship between climate warming index WI1990–2016 variations in the HT-HW countries, and AD/D (panel **A**), ALS/MND (panel **B**) and PD (panel **C**) prevalence. Each dot represents one of the 25 HT-HW countries considered.

**Figure 4 ijerph-19-13429-f004:**
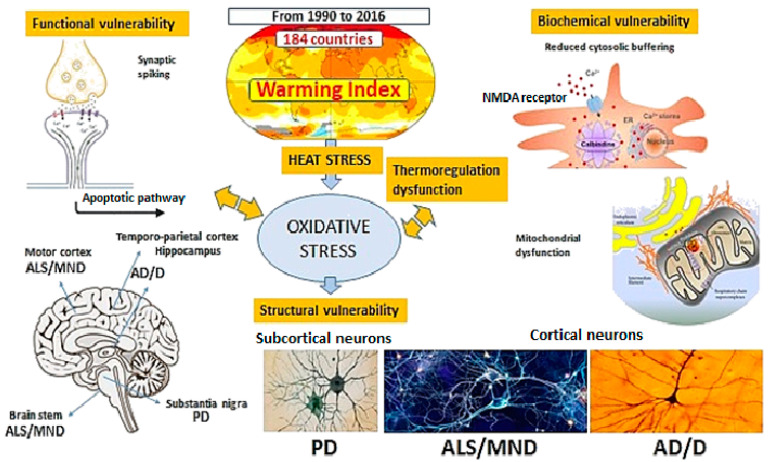
Scheme of the main heat stress-related types of neuronal vulnerability in neurodegenerative disease. Excessive synaptic spiking increases Ca^++^ into the neurons; Calbindin modulates Ca^++^ cytoplasmic levels; Prohibitin regulates mitochondrial function; cell structural differences influence neuronal susceptibility to oxidative stress.

**Table 1 ijerph-19-13429-t001:** Prevalence median values 1990–2016 variation for AD/D, ALS/MND and PD patients.

Prevalence	HT-LW	HT-HW	LT-HW	LT-LW
*AD/D*	−0.031	−0.031	−0.016	−0.040
*ALS/MND*	0.062	0.044	0.065	0.071
*PD*	0.152	0.215	0.122	0.157

**Table 2 ijerph-19-13429-t002:** Differences between the distributions of prevalence indices for the 4 clusters for (**A**) AD/D, (**B**) ALS/MND and (**C**) PD patients. The Table reports the *p*-value of the Student’s *t*-test calculated between the distributions of prevalence for the different clusters. Bold values identify the statistically significant correlations with Student’s *t*-test *p* < 0.05 (5%). For example, in (**A**) the fact that *p* = 35.5% for the distributions of the HT-HW and the HT-LW clusters, indicates that there is a 35.5% chance that the two distributions come from a different overall population.

(A) AD/D Prevalence	HT-HW	HT-LW	LT-HW	LT-LW
HT-HW	100.0%	35.5%	81.0%	6.4%
HT-LW		100.0%	63.4%	0.1%
LT-HW			100.0%	11.3%
LT-LW				100.0%
**(B) ALS/MND Prevalence**	**HT-HW**	**HT-LW**	**LT-HW**	**LT-LW**
HT-HW	100.0%	8.2%	0.8%	0.5%
HT-LW		100.0%	0.3%	0.3%
LT-HW			100.0%	88.8%
LT-LW				100.0%
**(C) PD Prevalence**	**HT-HW**	**HT-LW**	**LT-HW**	**LT-LW**
HT-HW	100.0%	0.0%	4.1%	0.1%
HT-LW		100.0%	86.7%	13.3%
LT-HW			100.0%	47.0%
LT-LW				100.0%

## Data Availability

Data collected in this study are all contained within the article and its Appendix A.

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
