# Peer review of "Effects of Global Warming on Patients with Dementia, Motor Neuron or Parkinson’s Diseases: A Comparison among Cortical and Subcortical Disorders"

_ijerph, 2022, doi:10.3390/ijerph192013429_

Round 1

Reviewer 1 Report

Manuscript ID.: ijerph-1937767

Title: Effects of global warming on patients with Dementia, Motor Neuron or Parkinson’s diseases: a comparison among cortical and subcortical disorders.

The topic is really interesting. However, I believe author should go through the comments and modify manuscript accordingly so as to get possible chance of its acceptance

Major Comments

Comment 1.  It is more appreciative if author will make some graphical representation / diagram specifically representing the overwhelming oxidative stress caused by global warming and possibility of occurrence of various neurodegenerative disorders as mentioned

Comment 2. Enlist the latest articles/findings dealing global warming and dementia related disorders

Comment 3: The manuscript has a lot of abbreviations. Please use a list of abbreviations that will help to keep up with the abbreviations used in the manuscript.

Comment 4:  Grammatical and punctuation errors should be rectified. Check the manuscript very thoroughly.

Comment 5:  Cite more important findings related to global warming in connection with dementia.

Comment 6: Discussion may be written in a better way

Reviewer 2 Report

The manuscript present a research article on the correlation between global warming and prevalence of patients affected by neurodegenerative disease.

Overall the article is well presented, the style and the order of section and subsection are clear, the English is well curated, and the effort of the authors is highly valuable. 

And moreover, I greatly appreciate the fact that all data are available in the manuscript (it is quite rare).

However some major issues emerged after reading the manuscript, as well as some statistical concerns.

MAJOR ISSUES

Line 12: the sentence “exposure to global warming… can lead to an increase in prevalence of neurological diseases” needs a citation (although not in the abstract, but at least in the main text) of an epidemiological study.

Line 36-39: briefly explain what you have found, instead of a simple autocitation.

Line 73: the authors included the “2016 average temperature”, defined as T2016. Why not including also the variable “1990 average temperature” (T1990) to perform the statistical analysis? A nation with higher temperature at the starting point (T1990) may be more prone to develop a change in prevalence (or not) of the diseases included in the study.

Line 73-77: please add a formula that better define the variables T2019 and WII1990-2016. The first is the “annual mean” temperature: measured everyday? The second is the slope of the monthly temperature: the mean temperature of each month (like it is supposed reading Line 83)? Further details are required.

Line 83-88: the authors choose to cluster the countries according to two parameters, according to the sample medians. Although it is not an error, it needs to be justified. Clustering data can be performed with much more sophisticated methods (k-means, Mean shift clustering, Hierarchical Clustering, etc.). The choice of a method should be justified.

Line 140-141: the authors affirm that “a linear relationship… was found for the HT-HW cluster only”. Does this mean that for the other clusters, different kind of relationship were found, like Quadratic, cubic, exponential, logarithmic?

Line 146-148: the sentence in basically a conclusion, and should be removed from the “results” section.

Line 194-217: the entire paragraph is quite long and difficult to read. It should be shortened. Further, it is not clear what is the relationships between the explained mechanism of calcium-toxicity and the global climate warming.

Line 218-222: this is an interesting hypothesis, but it is not supported by scientifically evidence. The body thermoregulation is controlled by several mechanisms, and although it is altered by the process of neurodegeneration, the authors should firstly prove that the brain temperature is altered as body temperature is altered in the neurodegenerative disease (or a citation is needed).

Line 227-239: similarly to the previous comments, this is an interesting hypothesis. However, the increased thermogenesis observed in PD people does not imply that the brain temperature is increased with respect of people without-PD. It should be the case that the body temperature of PD-people is higher, but the brain temperature is the same. A citation is needed before formulating such hypothesis.

No limitations of the study are included in the discussion of the manuscript.

MINOR ISSUES

Line 102-104: the reported numbers are the min and max? The Q1 and Q3? Also add the interquartile range.

Figure 1: add in the figure the names of each quadrants (HT-HW, HT-LW, etc.).

Figure 2: the abbreviation “prev=f(WI)” (prevalence as a function of warming index) should be explained in the footnotes of the figure.

Figure 2: add in the figure the slope and the significance of the slope.

A figure representing the relationships between prevalence and warming index for the other 3 clusters should be added (not only for the cluster HT-HW), eventually as supplementary material.

A figure representing the world map with nation highlighted according to the 4 clusters should be added.

Figure 3: where the three specimens depicted comes from? Personal images?

Line 240: change “suggestion” with “observation”.

STATISTICAL NOTES

Line 107: the observation that “distribution tends to be skewed towards lower values” is based on the visual examination of figure 1. However it should be better to use a test for skewness as the third standardized central moment of the random variable of the probability distribution (Omnibus K2 and Jarque — Bera).

Line 89-99: the authors compares epidemiological indices between cluster (obtained with climate indices) using a t-test. After an extensive literature search, I personally did not find any other research implementing such algorithm (except for the previously published work by the same group (https://doi.org/10.1016/j.joclim.2022.100130). Therefore I’m not sure that such method is statistically correct. First of all, a normality check should be performed before t-test. Further, since there are multiple comparison, a (Bonferroni or others) correction of the p-value should be performed. Finally, interpreting the p-value of a t-test as a percentage of “correlation” is not statistically sound, since “correlation” imply a completely different approach. Assuming that t-test is the correct choice (but remember the problem of multiple comparison), the percentage interpretation of p-value as a measure of correlation needs an extensive clarification.

TYPOS

Line 50: it is not clear what “home” stands for.

Line 200: change “Da” with “DA”.

Round 2

Reviewer 1 Report

The authors have inserted appropriate and suggested corrections.

I recommend the above manuscript for publication

Author Response

Thank you so much for your valuable comments.

Reviewer 2 Report

The authors answered to all my previous comments and the manuscript has been substantially improved, although I have some concerns regarding the sentence in the main text and table 2, where it is stated that "the p-value of Student's t-test calculated between the distributions of prevalence for the different clusters... the statistical significant correlations with Student's t-test p<0.05".

I leave the final decision to the editor.
